



# Arecibo measurements of D-region electron densities during sunset and sunrise: implications for atmospheric composition

Carsten Baumann[1], Antti Kero[2], Shikha Raizada[3], Markus Rapp[4,5], Michael P. Sulzer[3], Pekka T. Verronen[2,6], and Juha Vierinen[7]

[1]Deutsches Zentrum für Luft- und Raumfahrt, Institut für Solar-Terrestrische Physik, Neustrelitz, Germany
[2]Sodankylä Geophysical Observatory, Oulu University, Sodankylä, Finland
[3]National Astronomy and Ionosphere Center, Arecibo Observatory, Arecibo, Puerto Rico
[4]Deutsches Zentrum für Luft- und Raumfahrt, Institut für Physik der Atmosphäre, Wessling, Germany
[5]Meteorologisches Institut München, Universität München, Munich, Germany
[6]Space and Earth Observation Centre, Finnish Meteorological Institute, Helsinki, Finland
[7]UiT The Arctic University of Norway, Department of Physics and Technology, Tromso, Norway

**Correspondence:** Carsten Baumann (carsten.baumann@dlr.de)

**Abstract.** Earth's lower ionosphere is the region where terrestrial weather and space weather come together. Here, between 60 and 100 km altitude, solar radiation governs the diurnal cycle of the ionized species. This altitude range is also the place where nanometersized dust particles, recondensated from ablated meteoric material, exist and interact with free electrons and ions of the ionosphere. This study reports electron density measurements from the Arecibo incoherent scatter radar being performed during sunset and sunrise conditions. An asymmetry of the electron density is observed with higher electron density during sunset than during sunrise. This asymmetry extends from solar zenith angles (SZA) of 80 to 100°. This D-region asymmetry can be observed between 95 and 75 km altitude. The electron density observations are compared to the one-dimensional Sodankylä Ion and Neutral Chemistry (SIC) model and WACCM-D, a GCM incorporating the SIC ion chemistry. Both models also show a D-region sunrise/sunset asymmetry. However, WACCM-D compares slightly better to the observations than SIC especially during sunset when the electron density gradually fades away. An investigation of the electron density continuity equation reveals a higher electron ion recombination rate than the fading ionization rate during sunset. The recombination reactions are not fast enough to closely match the fading ionization rate during sunset resulting in excess electron density. At lower altitudes electron attachment to neutrals and their detachment from negative ions play a significant role in the asymmetry as well. A comparison of a specific SIC version incorporating meteoric smoke particles (MSPs) to the observations revealed no sudden changes in electron density as predicted by the model. However, the expected electron density jump (drop) during sunrise (sunset) occurs at 100 ° SZA when the radar signal is close to the noise floor, making a clear falsification of MSPs influence on the D-region impossible.

## 1 Introduction

The D-region is not only the lowest part of the ionosphere but also the faintest with its low abundance of free electrons. Only few measurement techniques allow investigations of this peculiar ionospheric region, i.e. rocket borne in situ measurements,





interpretation of VLF radio wave reflections and its sensing by means of incoherent scatter from free electrons and Faraday rotation. The latter technique was performed with the Arecibo incoherent scatter radar (ISR) in Puerto Rico from 1963 until December 1st 2020 with its large 305 meter dish and 2.5 MW radio wave transmitter (e.g. Isham et al., 2000, and references therein). This work aims to report specific sunset and sunrise D-region measurements performed with this one of its kind radar

during the end of August 2016.

The transmitted electromagnetic radar wave of the ISR is scattered from the ionospheric plasma. The detected backscattered signal can be described with Thomson scatter theory (Tanenbaum, 1968; Evans, 1969), which is adjusted for the collisional D-region plasma (Mathews, 1978). ISR measurements of the D-region have a long history and reach back to the beginning of the operation of High power Large Aperture radars like in Arecibo or elsewhere (e.g. Mathews et al., 1982; Kudeki et al., 2006;

Raizada et al., 2008; Kero et al., 2008).

First ISR investigations of the D-region ionosphere especially during sunset and sunrise were performed by Trost (1979). However, they did not investigate the differences between sunset and sunrise in detail. Other methods include medium frequency (MF) radar (e.g Coyne and Belrose, 1972; Li and Chen, 2014) and radio propagation methods (e.g. Laštovička, 1977) have also investigated the D-region during these times and discovered an asymmetry in the observed electron densities.

These observations led to further studies that investigated the interaction of the D-region with the background atmosphere. While Mathews et al. (1982) found gravity wave activity within the electron density measurements, Forbes (1981) investigated the influence of tides on the D-region ion chemistry based on the temperature dependence of reaction coefficients. The role of positive ion chemistry and its dependence on solar zenith angle and temperature plays a role during times of low ionization (Forbes, 1982). The importance of the neutral atmosphere has also been identified from diurnal variations of the temperature

to neutral density quotient inferred from spectral width of ISR signals (Ganguly, 1985).

Satellite observations of nitric oxide, i.e., the main ionized specie in the D-region (Nicolet and Aikin, 1960), show a distinct asymmetry in the NO concentration during sunset and sunrise (Siskind et al., 1998) as well. Friedrich et al. (1998) investigated these satellite result with respect to the D-region electron density and concluded that diurnal NO variations should be investigated within ionospheric models. Also atomic oxygen plays a prominent role in the lowermost D-region and underlies a

diurnal cycle to be taken into account for ionospheric modelling (Siskind et al., 2015).

Finally, ISR spectra from the lower ionosphere depend not only on the number of free electrons but also on the composition of the ions and abundance of charged aerosols. One peculiarity of the D-region is the possibility that negative ions can exist. Another one is the co-existence of the plasma with so called meteoric smoke particles which recondense from ablated meteoric material (Hunten et al., 1980). Cho et al. (1998) postulated a modification of ISR spectra due to the presence of heavy negative

charge carriers. A later measurement campaign reported in Strelnikova et al. (2007) successfully measured D-region ISR spectra that could be explained with the presence of negatively charged meteoric dust particles with a mean radius of around 1 nm. The existence of charged MSP dust has also been proven by means of rocketborne dust detections (e.g. Rapp et al., 2012; Robertson et al., 2013). This type of charged dust measurements including electron and positive ion measurements revealed that negatively charged dust influences the charge balance within the nighttime D-region (Friedrich et al., 2012). Modelling of

the D-region later confirmed this finding of Friedrich et al. (Baumann et al., 2013; Plane et al., 2014; Asmus et al., 2015). A





comprehensive review on the lower ionosphere that covers its complexity in full breadth has been published by Friedrich and Rapp (2009).

The scope of this work is to interpret the sunset and sunrise electron density observations with the help of modern ionospheric models. The measurements are compared to the Sodankylä Ion and neutral Chemistry (SIC) model (Turunen et al., 1996), a one dimensional model, and WACCM-D (Verronen et al., 2016), a global circulation model that includes a subset of the SIC ion chemistry scheme. By doing so, it is possible to distinguish between dynamical drivers and the pure ionospheric processes on the observed D-region asymmetry.

A further aspect of this study is to identify the expected impact of MSPs on the electron density during sunset and sunrise based on earlier model results (Baumann et al., 2015). Electrons effectively attach to MSPs when the D-region is in darkness, resulting in a sudden decrease of free electrons after sunset. The opposite occurs during sunrise when the Sun starts to shine on D-region altitudes, big amounts of electrons are then photo detached from negatively charged MSPs. The electron density measurements are expected to pin down if MSPs are actually an effective sink of electrons during unilluminated times.

The study is structured as follows. The Arecibo ISR measurements of the electron density are presented in Sect. 2. Section 3 compares these measurements with results from the SIC and WACCM-D model. The observed D-region asymmetry is analyzed in Sect.3.1. The results of the analysis are discussed in Sect. 4 and the conclusions are summarized in Sect.5.

## 2 Arecibo D-region measurements

The Arecibo radar consisted of the 305 m spherical antenna and a 430 MHz transmitter fed by a klystron RF amplifier. Its peak transmit power of up to 2.5 MW together with its high antenna gain of 61.1 dBi makes the Arecibo facility the most sensitive ISR in the world. The radar experiment was specially tailored for measuring the D-region electron densities. As a consequence a good measure of the background noise is crucial as it has to be subtracted from the backscattered power. Finally, the power profiles were calibrated using a plasma line measurement.

The details of this D-region radar experiment are as follows: The power profiles are obtained using an 88 baud code with 176 $\mu$s RF pulse length. It is used with a 400 $\mu$s gate delay and 500 range gates with 2 $\mu$s gate width. That results in an altitude range from 60 to 600 km. The 'noise' measurement uses a 2 baud (+/-) code with 0.2 $\mu$s RF pulse length. By using the shortest input pulse length, the transmitter had no time to ramp up the power. As a consequence, the transmitted power was near zero enabling a dedicated noise measurement. The 88 baud power profile and noise measurement used a 10 ms inter pulse period, running in a sequence of five seconds each. The plasma line measurement was done using a coded long pulse sequence (Sulzer, 1986) with 440 $\mu$s RF pulse length. The upper plasma line frequency was recorded in the frequency range 5.5 to 9.5 MHz with 4.8 kHz resolution. This plasma line measurement was done for approximately 5 minutes before (after - for sunrise) the above described main experiment sequence. The plasma line measurements were possible down to approximately 120 km.

The measured plasma line frequency can be related to the plasma frequency (and consequently the local electron density) using the formalism of Yngvesson and Perkins (1968). The measured power profile is directly proportional to the electron density after subtraction of the noise and correction of the resulting signal for range and near field antenna gain effects (Breakall

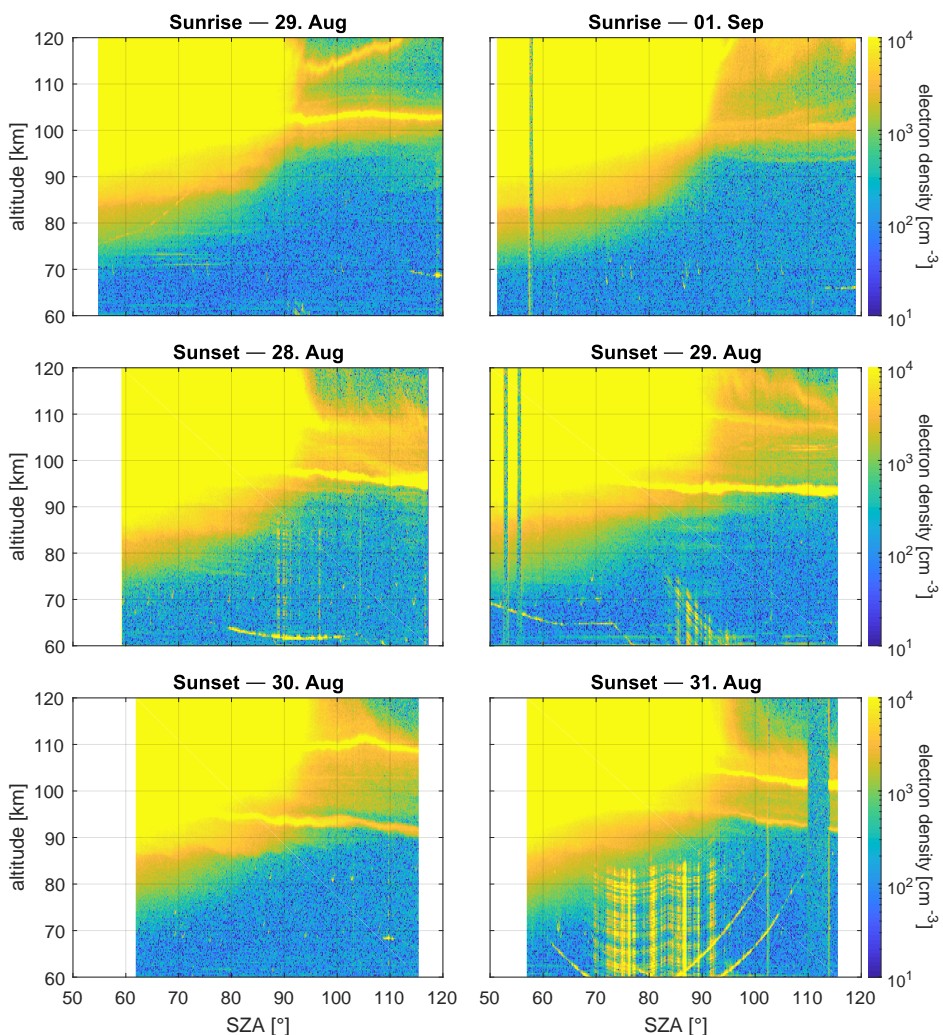

**Figure 1.** Arecibo radar measurements of the electron density from 28. August until 01. September 2016, four successful measurements during sunset and two during sunrise, x axis has been set to solar zenith angle (SZA) for better comparability.





and Mathews, 1982). This quantity is then calibrated with the measured electron density resulting in a calibrated electron
density profile from 60 to 600 km. We assume a constant calibration during the four-hour experiment period. Figure 1 shows
the result of this procedure after coherent integration of four sequences, resulting in 40s time resolution and 300 m altitude
resolution. The figure contains four sunset (28-31. August) and two sunrise measurements (29. August and 01. September).
Due to technical difficulties, the sunrise measurements on 29th and 30th of September were unsuccessful. The time axis of the
measured electron densities is transferred into solar zenith angle for a better comparison. The expected behavior of declining
electron density with SZA is visible in all altitudes. However, there are differences between the sunset and sunrise data.
Between 95 and 120 km sporadic E layers are present during nearly the whole measurement period (e.g. Hysell et al., 2009).
These layers are related to metal ion layers and atmospheric wind shears in these altitudes (e.g. Whitehead, 1961; Raizada
et al., 2011). Unfortunately, radio clutter occurs at lower altitudes with different severity as well. This originates from radar
beam side lobe reflections of airplanes and ships at these range gates.

To directly compare sunset and sunrise data, Fig. 2 shows measured electron densities at different altitudes from 95 km
down to 70 km as a function of SZA. The shown data represent the mean of the two sunrise and four sunset measurements.
For the case of the sunset dataset a 25% trimmed mean (e.g. Wilcox, 2011) is shown, doing that removes strong outliers due to
either sporadic E layers or low altitude interference. Furthermore, the shown lines represent the 20 point running mean and the
shaded regions indicate the standard deviation of this running mean. At low SZA the electron densities are remarkably similar
for sunset and sunrise. However, as the Sun reaches around 80° SZA sunset and sunrise measurements start to deviate.

At 95 km altitude the sunset electron density starts being higher than during sunrise at around 75 ° already. This asymmetry
remains in place for all SZA higher than that. However, this altitude region is likely influenced by the presence of sporadic E
layers that are very frequent in the evenings. The D-region asymmetry for 90 km starts at 85 ° SZA and also remains present
for all higher SZAs as well. For 85 km altitude, the asymmetry starts at 85 ° SZA as well. But the electron density values match
later at around 100° SZA again. At 80 altitude the D-region asymmetry is not so pronounced as in the altitude regions above
but also starts at 85 ° SZA and ends at 100°. The situation is more clear at 75 km altitude again. Here, the asymmetry already
starts at 80° and extends until 100 ° SZA. At 70 km a clear asymmetry cannot be observed anymore, because the signal to
noise ratio of the measurement is too low here.

The increasing standard deviation of the measurements indicates that the measured electron densities are close to or at the
noise floor of the Arecibo radar. The SZA at which the standard deviation sharply increases varies not only with altitude but
also with sunset or sunrise. The noise floor is reached at larger SZAs during sunset than during sunrise. This behavior indicates
a sunset/sunrise asymmetry of the ionosphere at altitudes from 90 to 75 km as well.

## 3    Comparison with ionospheric models

This section compares the electron density measurements to the Sodankylä Ion- and neutral-Chemistry model (SIC) (Turunen
et al., 1996) and WACCM-D (Verronen et al., 2016).





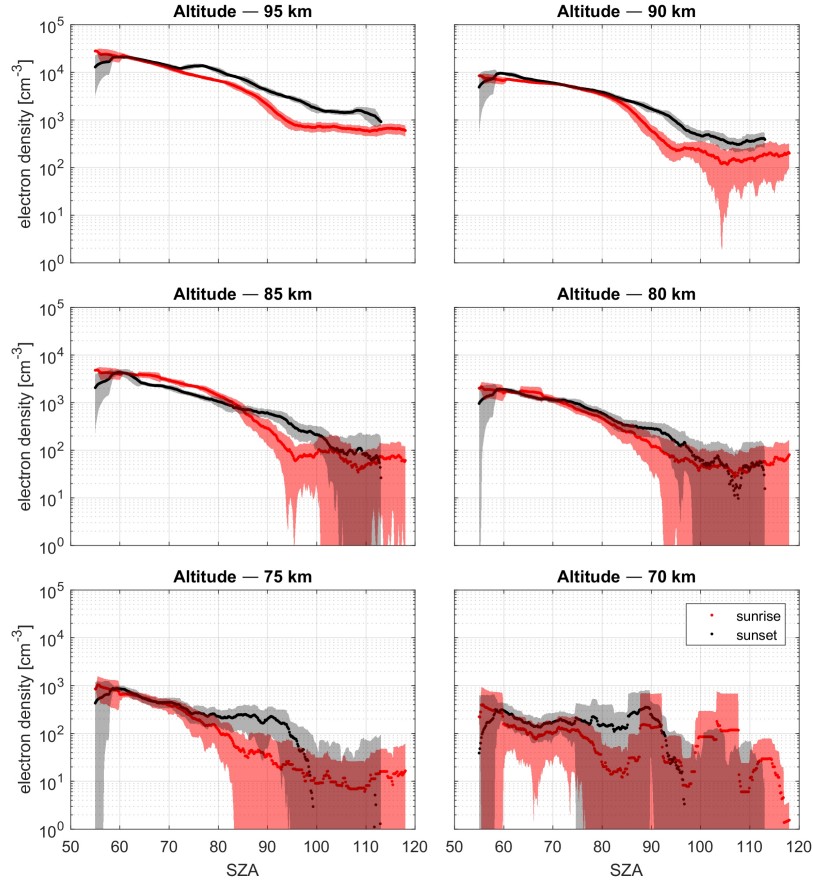

**Figure 2.** Sunset and sunrise comparison of the measured electron density at altitudes of 95 km down to 70 km as a function of solar zenith angle (SZA). Solid lines represent the 20 point running mean of two sunrise measurements and four sunset measurements. Shaded areas represent the 20 point running standard deviation. Sunset values represent a 25% trimmed mean to remove outliers due to sporadic E-layers.

We apply the SIC model in its original version and the version including meteoric smoke particles (Baumann et al., 2015). The SIC model is a one-dimensional ionospheric model designed specifically for the D-region with an altitude range from 20 to 150 km. This model has been widely employed across various applications, e.g. for polar energetic particle precipitation (e.g. Verronen et al., 2005) and as the model for inversion of electron density profiles from spectral riometry (Kero et al., 2014).

The SIC model includes a chemical scheme of 41 positive ions, 29 negative ions, and 34 neutral species to represent the D-region and the underlying mesosphere and lower thermosphere. The model takes into account ionisation processes from solar radiation, precipitating electrons and protons, and galactic cosmic rays. The chemistry scheme includes ion-neutral reactions, electron attachment/detachment, and electron-ion and ion-ion recombination. Vertical transport of some minor neutral species is represented by parameterized eddy and molecular diffusion. But there is no vertical transport of ionized species and 130    no horizontal transport because SIC is a 1D model. For a more comprehensive description of the SIC model, see Verronen



(2006). To represent meteoric smoke particles (MSPs) in SIC, a particle size distribution that is based on (Megner et al., 2006) was incorporated into SIC (this version will be called SIC-MSP from now on). To couple the neutral MSP to the D-region ionosphere, SIC-MSP derives the MSP charging rates. SIC-MSP handles direct electron and ion attachment to neutral MSPs as well as charged MSPs. The most relevant MSP related processes are the electron attachment to neutral MSPs and the consecu-

tive electron photodetachment of negatively charged MSPs induced by sunlight. The interplay of both processes is particularly interesting during sunset and sunrise, as during that time the charging and corresponding decharging of the negatively charged MSP fraction occurs.

The second model being compared to the measurements is the WACCM-D model (Verronen et al., 2016). This global circulation model is a variant of the Whole Atmosphere Community Climate Model (WACCM) that incorporates a D-region ion

chemistry scheme based on the SIC model, and includes 307 reactions of 20 positive ions and 21 negative ions. For comparison to the electron density measurements performed in 2016 we use a model results for the year 2005. By using WACCM-D data from the same season we arrive at similar SZAs, Ly-$\alpha$ fluxes and similar overall conditions despite introducing slight difference due to comparing measurement and model data from different solar cycles. In contrast to SIC, WACCM-D is able to handle the diffusion and transport of all species (neutral and ions), vertically as well as horizontally. However, neither model considers

thermospheric plasma transport due to electromagnetic forces or ambipolar diffusion. The WACCM-D model gives out data for the whole globe with a bin size of $1.9°x2.5°$ in latitude and longitude and a one-hour time resolution. For the following analysis we chose latitude $18°$ and made use of different longitude bins around the globe.

The SIC model is run for Arecibo radar's geographical location ($66,8°$ W, $18.3°$ N) and for the same time period as the observations. The altitudes covered by WACCM-D and SIC reach up to 150 km altitude, here we will concentrate on the

altitude region between 77 and 91 km.

Identical to the Arecibo measurement data, the time axis of the output of both models has been transferred into SZA. The SIC model's time resolution of 5 min transfers into approximately $1°$ SZA. The one hour time resolution of the WACCM-D model however, is much more sparse. To increase the SZA resolution of WACCM-D, output data from all longitudes covered in the spatial resolution are transferred to SZA. By doing so the resolution of the WACCM-D results can be reduced to below

$1°$ SZA. This handling is valid under the assumption that there are only minor longitudinal variations in the D-region. This is case in the low latitude D-region ionosphere which is solely governed by photoionisation.

In contrast to Fig. 2, the comparison of the electron density measurement with the ionospheric models is separated into sunset and sunrise conditions. Figure 3 shows the mean sunrise measurements as well as the corresponding model results on the left panels. The right panels of Fig. 3 show the sunset comparison of model results and electron density measurements.

The altitudes that have been chosen for comparison are 91 km, 85 km, 80 km and 77 km. These altitudes have been chosen because they closely match the pressure levels of WACCM-D. Measurements at higher altitudes are not compared to the used ionospheric models because these models do not fully cover E-region physics, like sporadic E-layers. Measurements at lower altitudes are often too close to or at the noise floor of the radar and are not considered for comparison.

The sunrise comparison in Fig. 3 at 91 km altitude shows that a good agreement between SIC/WACCM-D and the measure-

ment at lower SZA. However, the shape of the electron density rise during sunrise is not reproduced with the models. SIC and



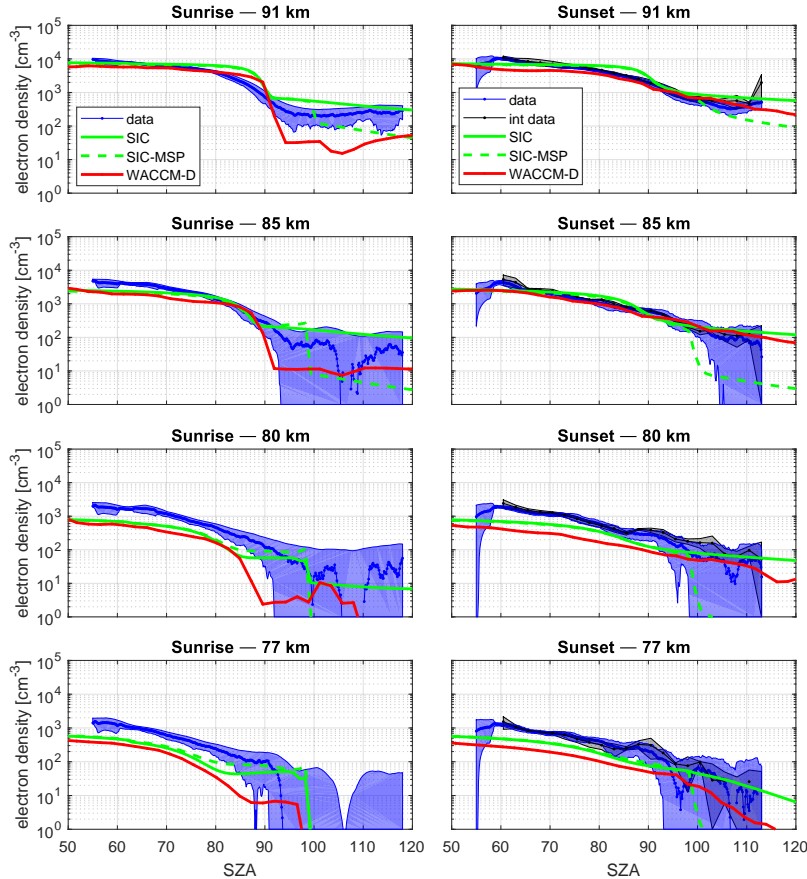

**Figure 3.** Comparison of measured electron densities with model results from the SIC and WACCM-D model for sunrise (left) and sunset (right) at 91 km, 85 km, 80 and 77 km altitude. The blue line is the 20 point running mean of the measurements and the blue area is the corresponding standard deviation. Green lines indicate SIC model in its standard version (solid) and with meteoric smoke particles incorporated into the ion chemistry (dashed). WACCM-D results are given in red.

WACCM-D expect a relative sharp electron density increase, while the measurements indicate a prolonged electron density increase for an extended period. The expected electron density jump of the SIC-MSP model at 100° SZA is not observed.

The sunset comparison in Fig. 3 at 91 km altitude shows good agreement between ionospheric models and the measurements. However, at SZAs < 70 ° WACCM-D slightly underestimates the electron density. The SIC model overestimates the electron density between 80 and 90 ° SZA. WACCM-D reproduces the shallow electron decrease during sunset slightly better than the SIC model. The sudden drop of electron density in SIC-MSP run is within one standard deviation of the measurements for SZAs between 100 and 110°.

The measured sunrise electron density at 85 km is very well captured by SIC and WACCM-D as well. The early morning electron density (high SZA) in WACCM-D is however much lower compared to the SIC model. However, the measurement





standard deviation is high at SZA > 90°, which makes a distinction between the models impossible. At SZA greater than 90, the SIC model is at the upper edge of the measurements and WACCM-D at the lower edge. Moreover, the SIC-MSP results remain feasible as they repeatedly lie within the standard deviation of the observation.

During sunset at 85 km WACCM-D compares best to the slowly decaying electron density measurements. The SIC model has a slightly steeper electron density drop between 85 and 95° SZA, but also shows generally a shallower electron density
decay in contrast to the steeper electron rise during the morning hours. The SIC-MSP results are not fully resembled by the standard deviation of the measurement, and a distinct electron density drop is not visible as well.

When going down to lower altitudes like 80 and 77 km SIC and WACCM-D underestimate the number of free electrons. During the sunrise, WACCM-D still rise from very low electron density. The SIC results show higher nighttime values of electron density, however the increase occurs at later at smaller SZAs. The standard SIC as well as the SIC-MSP version show
a distinct jump in electron density at around 100° at both altitudes. WACCM-D shows this jump only at the 77 km altitude. These electron density jumps are within the standard deviation of the measurement. However, the mean value do not show this electron jump at 80 km altitude, but at 77 km slightly shifted to a smaller SZA of 95°.

During sunset the models underestimate the electron density compared to the measurements at altitude 80 and 77 km, WACCM-D shows even lower values than SIC. However, both models represent the slow electron density depletion during
sunset. WACCM-D produces a slightly smoother decay at 80 km than SIC for SZA < 90°. At higher SZA the electron density in WACCM-D decays faster than in SIC, but both models are within the standard deviation of the electron density measurement. The electron density drop of the SIC-MSP is within the standard deviation again, but is not indicated from the mean measured electron density.

## 3.1 D-region asymmetry

This section investigates the observed D-region asymmetry during sunset and sunrise (60° < SZA < 100°) with the help of SIC and WACCM-D. The investigation concentrates on the ionospheric processes being implemented within these models and how they behave during sunset/sunrise.

The continuity equation of the electron density is central for the description of the ionosphere. It is rather complex in the D-region as also negative ions can exist:

$$\frac{d[N_e]}{dt} = q - \alpha[N_e][N_{I^+}] - \beta[N_e][N_n] + (\gamma[N_n] + \gamma_p)[N_{I^-}]. \tag{1}$$

Here, $q$ is the electron production by ionization and $\alpha[N_e][N_{I^+}]$ is the electron loss due to electron recombination with positive ions. The loss term $\beta[N_e][N_n]$ describes the electron attachment to neutrals ($[N_n]$), which is important at altitudes below 80 km. Oppositely, the term $(\gamma[N_n] + \gamma_p)[N_{I^-}]$ is the electron detachment from negative ions. This process can be divided into the collisional electron detachment ($\gamma$) and electron detachment by solar photons ($\gamma_p$). The continuity equation above lacks the
transport term of the electron density. Direct plasma transport is not considered in SIC and therefore cannot be discussed in this study.





For further analysis Fig.4 shows all terms of the continuity Eq. 1 during sunrise and sunset for altitudes 91 km, 85 km, and 80 km. The results are based on the SIC model in its standard version without MSP. Given values represent the sum of all individual reaction rates for each term of Eq.1, i.e. the product of reaction rate coefficient with the appropriate concentrations of the reaction partners.

At 91 km altitude the ionization rate dominates during both sunrise and sunset conditions for SZAs up to 100 °. While the electron - positive ion recombination rate follows closely the ionization rate during sunrise, this is not the case during sunset between 90 and 100 ° SZA. Here, electron - positive ion recombination rate falls off slower than the ionization rate. At even larger SZA the electron attachment to neutrals and electron detachment from negative ions dominate the continuity equation.

The situation is similar at 85 km. Here as well, ionization rate and electron ion recombination do not match during sunset. However, the electron attachment to neutrals and detachment from negative ions start to be relevant already. These processes related to negative ions show a asymmetry between sunrise and sunset as well.

At 80 km altitude the situation becomes different. During sunrise the photo induced electron detachment from the negative ion reservoir occurs at 100°. This process dominates until this reservoir is emptied, after that the collisional electron detachment is dominant again. The electron ion recombination rate still falls off slower than the ionization rate during sunset but the recombination rate also rises slower than the ionization rate during sunrise. But both these processes fall behind the rates of electron attachment to neutrals and electron detachment from negative ions at all times. This results in an asymmetry of the electron density in SIC because both processes show distinct patterns during sunrise and sunset.

## 4  Discussion

The D-region sunset/sunrise asymmetry is a phenomenon that has been studied for several decades with various techniques. The asymmetry is usually characterised by MF radars measuring the transmitted wave's Faraday rotation (Coyne and Belrose, 1972) and by oblique radio link amplitudes at different MF frequencies (Laštovička, 1977). We report the first direct measurements of the D-region electron density asymmetry by calibrated incoherent scatter radar observations.

Here, the D-region electron density during sunrise and sunset is specifically observed by means of ISR radar in Arecibo (Puerto Rico). The observations show significantly lower electron densities during morning hours compared to evening hours, when considering SZAs between 80 and 100°. For lower SZAs the electron densities do not differ significantly. This asymmetric behavior is observed for altitudes between 90 and 75 km altitude.

MF radar observations usually show asymmetries in the observed electron density already starting at SZAs of 40° (Li and Chen, 2014), however they tend to observe at even lower altitudes. The D-region asymmetry has also been observed by means of VLF observations and these observations also indicate a D-region asymmetry starting at lower SZAs compared to the findings presented in this study. The reason for different observations of the asymmetry remains unclear and is left to be investigated in future studies.

In this study we also conduct a comparison of time-dependent ionospheric models with the measured electron densities. The one-dimensional SIC model and three dimensional WACCM-D have been used to model the sunset and sunrise electron density.



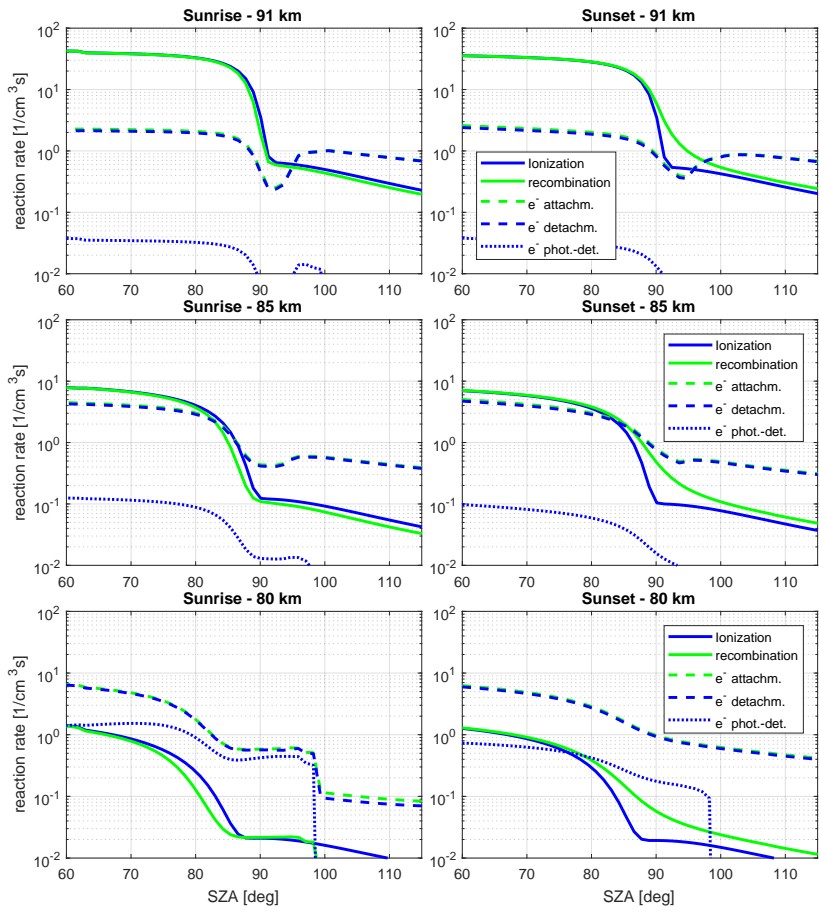

**Figure 4.** Components of the electron continuity equation 1 for altitudes 91 km (top), 85 km (middle) and 80 km (bottom) for sunrise (left) and sunset (right). The components are the ionization rate (blue-solid), electron-positive ion recombination rate (green solid), electron attachment to neutrals (green-dashed) and electron detachment from negative ions (blue-dashed). The latter is the sum of collisional detachment (not shown) and photo detachment of electrons from negative ions (blue-dotted).





Both models employ similar ionospheric reaction schemes. Therefore, SIC and WACCM-D show similar results (cf. Fig. 3) but in the end WACCM-D agrees better with the observations especially during sunset. The advantage of WACCM-D lies in being a general circulation model. The neutral background of SIC is provided by the NRLMSIS model (Picone et al., 2002) that is a climatological model of the upper atmosphere. The difference between both models has to originate from transport or the background atmosphere's temperature and its representation within either model. For instance, tides can impact the ion

chemistry significantly and alter the abundance of heavy water cluster ions (e.g. Forbes, 1982). A thorough analysis of the differences between SIC and WACCM-D especially in the ionosphere at low latitudes is subject to a future study.

The performance of both models in comparison to the observations is not so good during sunrise conditions. That can be a result of unknown reaction rate coefficients for electron detachment from some negative ions. Not all negative ions have a direct reaction path to lose electrons but require a detour transfer reaction to a negative ion specie that actually can lose electrons.

The analysis of the electron continuity equation for the SIC model (cf Sect. 3.1) reveals the underlying processes of the observed D-region asymmetry. At altitudes of 85 and 90 km altitude the interplay between electron ion recombination rate and ionization rate is most important. During sunset the recombination rate is higher than the ionization rate, but during sunrise both rates match closely. The difference between both rates during sunset can be explained by the fast declining ionization rate due to Ly-$\alpha$ atmospheric absorption as the Sun goes down and the inability of the recombination reactions to follow with the

same speed. The remaining electrons and positive ions just need additional time to recombine and reach a steady state with the lower ionization rate later during the night (SZA $> 100°$). At lower altitudes the electron attachment to neutral species and detachment from negative ions are more important and dominate the shape of the asymmetry within SIC. A detailed analysis of these time-dependent processes and an identification of involved ion species especially for WACCM-D is subject of a future study.

The presence of charged meteoric smoke particles (MSPs) has been proven by several rocket-borne and radar observations. These chargeable MSPs are expected to cause distinct jumps and decreases of electron density during sunrise and sunset at D-region altitudes (SZA = 100°). A thorough analysis of this experiment, however, does not show these distinct features in the electron density (cf. Sec. 2). However, the comparison of the observation to the SIC-MSP model Baumann et al. (2015) shows that these features occur during times when the sensitivity of this experiment is not sufficient to test our understanding of MSP

effects.

## 5 Conclusions

In this study, we concentrated on the sunset and sunrise behavior of the D-region ionosphere and measured the electron density with the Arecibo incoherent scatter radar located in Puerto Rico. A sunset/sunrise asymmetry of the electron density has been observed with ISR technique for the first time. These observations have been compared to the 1D ionospheric model SIC and

the 3D GCM WACCM-D that has the SIC ion chemistry included.

The identified asymmetry in the D-region electron density is a higher electron density during sunset than during sunrise for the same SZAs. This asymmetry was observed for SZAs greater 80 ° and in an altitude region between 75 and 95 km. Other



studies using MF radar and VLF observations (Coyne and Belrose, 1972; Laštovička, 1977; Li and Chen, 2014, e.g.) reported this D-region asymmetry for lower SZA (down to 40 °) and lower altitude regions. The present ISR observation showed that the observable time span of the D-region asymmetry decreases with altitude and shifts to higher SZAs.

The observed D-region asymmetry was analyzed by comparison to the 1d ionospheric model SIC and the 3D GCM WACCM-D that also includes a similar ion chemistry scheme. Both models, SIC and WACCM-D, show signatures of an asymmetry between sunset decline and sunrise growth of electron density. However, WACCM-D generally reproduces the observed D-region asymmetry better. An analysis of the continuity equation of the ionospheric electron density showed that SIC's asymmetry originated from a higher electron ion recombination rate than the ionization rate during sunset. As the Sun goes down, the electron-ion recombination is not fast enough and needs time to reach a steady state with the rapidly declining ionization rate. At an altitude of 80 km and below, the electron attachment to neutrals and electron detachment from negative ions govern the shape D-region electron density during sunrise and sunset here. The differences between SIC and WACCM-D could be attributed to the vertical and horizontal transport processes being taken into account in WACCM-D but not in SIC, while the ion chemistry scheme is similar in both models. It is very likely that the background neutral atmosphere, its temperature and dynamics, play a significant role in the D-region ionosphere during times of weak ionization and should be further investigated in the future.

In addition, the D-region observations did not clearly indicate a sudden electron density increase/depletion caused by decharging/charging of MSPs during sunrise/sunset as indicated by specific ionospheric modelling (Baumann et al., 2015) at a SZA of 100°. However, the ISR measurements during these high SZAs lack sensitivity at altitudes below 90 km. The lack of signal power increased the uncertainty of the measured electron density making an ultimate conclusion impossible or at least ambiguous. Further studies on the optical and charging properties of MSPs and further D-region observations during different times throughout the day remain necessary.

*Code and data availability.* The raw radar data (power profiles and plasma line measurements), processed data, as well as plotting routines for Fig. 3 and 4 have been made available on Zenodo (Baumann et al., 2022).

*Author contributions.* The research idea was conceived by CB, AK and MR. The radar experiment was conducted by MPS. Data analysis was performed by CB with support from AK, SR, MR and JV. WACCM-D data was provided by PTV. Interpretation of the results was performed by CB, AK, PTV and MR. All authors contributed to the writing of the manuscript.

*Competing interests.* The authors declare that there are no competing interests present.





*Acknowledgements.* The authors thank Nestor Aponte and Phil Perillat for their support at the radar site. The radar observation itself was funded at the time by NAIC/NSF/SRI based on proposal T3087. The work of A.K. is funded by the Tenure Track Project in Radio Science at Sodankylä Geophysical Observatory/University of Oulu. The work of P.T.V. is supported by the Academy of Finland grant no. 335555 (ICT-SUNVAC).



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
