# Peer review of "Arecibo measurements of D-region electron densities during sunset and sunrise: implications for atmospheric composition"

_Annales Geophysicae, 2022_

## Author Comment (AC1)

Response to Reviewer #1

**Arecibo measurements of D-region electron densities during sunset and sunrise: implications for atmospheric composition**

Carsten Baumann, Antti Kero, Shikha Raizada, MarkusRapp, Michael P.Sulzer, Pekka T.Verronen, JuhaVierinen

Thank you very much for your review of our manuscript and for the very positive appreciation of our work. We have taken all your comments into account and will answer them point by point in the following.

*(1) I couldn't find any statement about the solar and geomagnetic conditions during the measurements, only when it comes to using GCM 2005 data given an equivalent solar condition.*

The referee is correct, geomagnetic and information n the suns activity is missing. We add the following passage to the manuscript within Sect. 2: "Geomagnetic activity during the measurement period was low to moderate with Kp index ranging from 0 to 4. The DST index reached a minimal value of -57 nt on September 1st 2016 10:00 UTC at the very end of measurement campaign. This enhanced geomagnetic activity The activity of sun was moderate with radio flux F 10.7 ranging between 80 and 100 sfu. The strongest solar flare was of type C2.2 and occurred on August 31st 20:19 UTC (GOES), but no immediate impact on the D-region is visible in the data."

*2) L19: electron density measurement techniques are introduced: in situ, VLF radio wave reflections, ISR measurements. Later on, L33, suddenly MF radar techniques are mentioned if not highlighted as it is in the discussion section. I'm confident the MF techniques, given the system is well capable of it, is more useful and reliable than inferring VLF radio wave propagations...? Perhaps MF techniques could be mentioned already in L21?*

The reviewer points to a confusion within the introduction section. The introduction has been reorganized, so that it represents the MF radar capabilities in a better way. We also include a reference of a review on MF radar techniques [Reid, 2015].

*3a) Fig1: I suggest to adjust the color scales to higher electron densities max. 5e4 or 1e5 to limit the saturation for the E-region peak, even though it's not in the focus of this paper. But it will beautify the plot. 3b) Fig1: I assume the obvious gaps have been excluded for the subsequent statistics, but couldn't find a note? 3c) Fig1: Judging on that plot the noise floor, so the sensitivity, is near 10-100 $el/cm^-3$. Especially with densities below 10, I'd be very careful near that noise floor... From Fig2 and Fig3 it doesn't look like you applied a kind of SNR selection, do you?*

The reviewer indicates that Figure 1 in the manuscript uses a color scale that is to some extent ill-suited for the E-region heights. Figure 1 has been replotted with the suggested color scale, see Figure 1. However, electron density structures at low densities are less pronounced with this color scale. Therefore, we will stick to the original color scale as it highlights regions and times with lowest electron densities which are, as you point out as well, in the focus of this paper. We add a sentence to the caption of the figure, stating that the color bar has been intentionally set like this.

[Figure]

Figure 1: As in the manuscript but with $10^1$ to $10^5$ cm$^{-3}$ color scale for the measured electron density.

Data gaps are excluded by applying the trimmed mean for the sunset observations. For the sunrise measurements, data gaps did only occur on 31. August very early in the morning, when the sun was still far below the horizon not affecting the analysis.

You are correct, we have not applied a SNR selection method, and you are also right that the sensitivity of the measurement is around 10 - 100 $el/cm^-3$. That is also mentioned within the text of the manuscript as well. The referee is also right, that Fig. 2 and 3 of the manuscript show electron densities scales down to $1el/cm^-3$, that has been only done to help guide the eye and indicate low or absent signal. We do not draw any conclusions from these values.

*4) L102: A "25% trimmed mean" is used to explicitely suppress sporadic E layer echoes. How good does this suppression work considering the echoes are pretty intense. Perhaps adding a plot with an example to the manuscript or only as a reply comparing to e.g. median? Do you apply the same method to suppress the airplane/ship clutter? @ L98*

Indeed, the description of the trimmed mean is not really clear. A 25% trimmed mean basically removes the greatest outlier before performing the mean. Given values are the mean of the 3 measurements which are closest to each other. This procedure helps removing not only sporadic E region electron densities but also data gaps and clutter from ships or planes. We have expanded the description for clarification. The sentence now reads: "For the case of the sunset dataset a 25% trimmed mean [e.g. Wilcox, 2011] is shown, doing that removes one strong outlier from the 4 observations either due to sporadic E layers, low altitude interference from ships/planes or data gaps during periods when the transmitter was off."

*5) L110: At 80 altitude... -¿ At 80 km altitude...*

Thank you, the missing unit has been added.

*6) L141: I agree the years 2005 and 2016 were quite similar talking about the solar activity. I guess for that purpose that's sufficient, but what about the dynamics? From my impression WACCM-D is nicely reproducing daily means at late summer for these altitudes, not that sure about the time scales you're looking to, though.*

The author is correct about the capabilities of WACCM-D. The scope of this paper is no detailed comparison of WACCM-D to the D-region observations. More specific model runs with lower time resolution are needed to fully assess the D-region sunset and sunrise with WACCM-D. That is subject of an upcomming paper.

*7) L152: Nice idea to use multiple longitudes to create a higher SZA resolution... I'd worry about horizontal transport effects (dynamics).?. 1° longitude corresponds to roughly 100km displacement.*

The reviewer is correct that the procedure to use data from different longitudes increases the time resolution but also limits the possibility to make statements on the horizontal transport. We have added a sentence to further clarify the issue. In order to assess the full dynamics within WACCM-D during sunset and sunrise model runs with higher time resolution are needed.

"The assumption is that the SZA-driven changes at sunrise/sunset, also on dynamics, are much stronger than any dynamical artifact coming from sampling different longitudes at the same time. Visual inspection of the WACCM-D data shows that no electron density artifacts are present."

*8) L240 (, L285 and somewhere earlier): "Both models employ similar ionospheric reaction schemes." I think that's not strictly correct as*

*you pointed out earlier SIC and WACCM-D incoporate different amount of pos./neg. ions, and thus also possible reactions. I suppose to relax it by "equivalent", but not similar.*

We incorporated the suggestion of the reviewer into the manuscript.

*"cosmetics": - consistent use of value and ° without a space, L106, L108, L109, L111, L112 - L232: ..."altitudes between 90 and 75 km altitude." -¿ remove the latter*

We thank the reviewer pointing out the issue with spaces between numerals and its units as well as the typo. Both errors have been corrected throughout the manuscript.

**References**

Iain Murray Reid. MF and HF radar techniques for investigating the dynamics and structure of the 50 to 110 km height region: a review. *Progress in Earth and Planetary Science*, 2(1), oct 2015. doi: 10.1186/s40645-015-0060-7.

Rand Wilcox. *Introduction to Robust Estimation and Hypothesis Testing*, chapter 2.2.3 The Trimmed Mean, page 32. Elsevier, 2011. ISBN 9780123870155.

---

## Author Comment (AC2)

Response to Reviewer #2

**Arecibo measurements of D-region electron densities during sunset and sunrise: implications for atmospheric composition**

Carsten Baumann, Antti Kero, Shikha Raizada, MarkusRapp, Michael P.Sulzer, Pekka T.Verronen, JuhaVierinen

We would like to thank the referee #2 for taking the time to review our manuscript. The constructive comments on our manuscript and the overall positive judgement of our work are appreciated. Especially, the additional VLF references are appreciated as this topic was not well represented.

In the following we will address all comments point by point.

*Page 1 Para 20: The relevant citations be added to rocket borne in situ measurements (citations), interpretation of VLF radio wave reflections (citations) and its sensing by means of incoherent scatter from free electrons and Faraday rotation. For the VLF following citations are suggested:*

*Han, F., & Cummer, S. A. (2010a). Midlatitude daytime D region ionosphere variations measured from radio atmospherics. Journal of Geophysical Research, 115, A10314. https://doi.org/10.1029/2010JA015715*

*Kumar, A., & Kumar, S. (2020). Ionospheric D region parameters obtained using VLF measurements in the South Pacific region. Journal of Geophysical Research: Space Physics, 125, e2019JA027536.*
*https://doi.org/10.1029/2019JA027536*

*Maurya, A. K., Veenadhari, B., Singh, R., Kumar, S., Cohen, M. B., Selvakumaran, R., et al. (2012). Nighttime D region electron density measurements from ELFVLF tweek radio atmospherics recorded at low latitudes. Journal of Geophysical Research, 117, A11308.*
*https://doi.org/10.1029/2012JA017876.*

*Thomson, N. R., Clilverd, M. A., & McRae, W. M. (2007). Nighttime D region parameters from VLF amplitude and Phase. Journal of Geophysical Research, 112, A07304. https://doi.org/10.1029/2007JA91227*

We incorporated your suggested references throughout the manuscript at the mentioned places.

*Page 12-13, Para 270: Other studies using MF radar and VLF observations (Coyne and Belrose, 1972; Laštovicka, 1977; Li and Chen, 2014, e.g.). None of the citation is from VFL study. Please check. The VLF is the most coseffective and forms a novel tool to study D-region under the normal and natural Hazards which I think needs to be given bit more emphasis.*

Thank you for pointing out that appropriate references were missing, we added studies from your suggested list. The citations at the location you mentioned have been

separated according to their subject (VLF and MF).

---

## Author Comment (AC3)

Response to Reviewer #3

**Arecibo measurements of D-region electron densities during sunset and sunrise: implications for atmospheric composition**

Carsten Baumann, Antti Kero, Shikha Raizada, MarkusRapp, Michael P.Sulzer, Pekka T.Verronen, JuhaVierinen

We thank the reviewer for assessing the quality of our manuscript. The effort to identify shortcomings and minor flaws is also highly appreciated. and for taking the time to review our manuscript. The constructive comments have been taken into account to further improve the manuscript quality.

In the following we will address all comments point by point.

*The authors should give the full name for WACCM-D and GCM in the abstract. In the text, full names should be given in the first place where the abbreviation appears.*

Thank you for the corrections, we changed the manuscript accordingly in the abstract and introduction section.

*Fig. 1: Is $10^4$ cm $^{-3}$ the maximum obtained value of electron density? I ask, because I have the impression that this value is given on large parts of the displayed graphs. I have impression that higher values were obtained but that they are seen as $10^4$ cm $^{-3}$ due to the limitations of the domains in the display.*

The reviewer is correct, that values above $10^4$ cm $^{-3}$ have the same color as $10^4$ cm $^{-3}$. This issue was also pointed out by reviewer #1. However, the color scale maximum of the figure has been intentionally set to $10^4$ cm $^{-3}$ in order to highlight the lower electron densities of D-region and lowermost E-region. The aim was also to show data from altitudes not being analyzed in detail later on, so that a more complete picture of the lower ionosphere is presented. The caption of figure 1 has been extended with a sentence, so that the choice of the color scale is justified.

*Lines 122-123: D-region heights is located between 50-60 km and 90 km. For this reason, the part "... the D-region with an altitude range from 20 to 150 km. " should be rewritten.*

The reviewer is correct that this sentence is ambiguous with respect to the D-region altitude range. The sentence has been split in two and now reads: "The SIC model is a one-dimensional ionospheric model designed specifically for the D-region. It covers the altitude range from 20 to 150 km including an ion chemistry for the most prominent ions."

*To my knowledge, the SIC model is primarily used for polar region analyzes. The authors should explain the possibility of applying this model*

*(its original version and the version including meteoric smoke particles) to the area observed in this study. Is it necessary to make some corrections (eg those related to the chemical composition, the influence of the magnetic field, etc.) in these versions of the model to make their application relevant to other areas, or changes depending on observed areas and observation periods can be made in the input files?*

The reviewer is correct, that the SIC model has been frequently used for polar latitudes. However, the ion chemistry scheme is not changed for the present study covering low latitudes. Only the SZA, photoionization, galactic cosmic rays are different at low latitudes. These differences can slightly change the resulting ion composition and of course diurnal electron density progression. SIC also solves for ozone-related chemistry, so that part of the neutral atmosphere responds as well. NRLMSISE-00 provides the major species depending on location and solar activity. The effect of the magnetic field on the ionosphere is not handled within SIC, as the D-region is a highly collisional plasma and currents do not play a significant role here. The only parameter that has been changed is the vertical eddy diffusion coefficient. For clarification we have added the following sentence: "SIC has been extensively used to model the high latitude ionosphere in combination with EISCAT radar observations. It's application to low latitude D-region like in Arecibo (Puerto Rico) however, does not need very specific changes. Photoionization and ionization due to galactic cosmic rays are calculated for the location in question. Of course, particle precipitation as ionization source is turned off and besides that only a slight adaptation of the vertical diffusion coefficient is needed. The individual ion species and involved ion chemistry remains untouched."

*Does the model use Eq. (1) for calculations of the effective values of the parameters related to the respective processes, or does it consider the reactions of a single type of particles (and consequently coefficients corresponding to these processes considered in particular)? The authors should explain this in the text. In case the first variant is applied, the names of the corresponding coefficients should be written and it should be explained how the corresponding effective coefficients are changed in accordance with the observed conditions. In case the second variant, Eq. (1) should be rewritten with sums and corresponding indexes and all these quantities should be explained in the text.*

The reviewer is indeed correct, that the formalism used in Eq. 1 is not fully clear. We have adopted the summation formalism and the equation now reads:

$$\frac{d[N_e]}{dt} = \sum_i q_i - \sum_j \alpha_j [e^-][I_j^+] - \sum_k \beta_k [e^-][N_k] + \sum_l \gamma_l [N_l][I_l^-] + \sum_m \gamma_m^p [I_m^-]. \quad (1)$$

In order to further clarify, the following sentence has been added: "The summations and their indices indicate that the ionospheric reactions (Verronen 2006) are handled with their corresponding reaction partners."

*line 204: $\gamma$ and $\gamma_p$ are the effective coefficients related to the collisional*

*electron detachment and electron detachment by solar photons, not the collisional electron detachment and electron detachment by solar photons.*

We incorporated the correction of the reviewer into the manuscript.